# Unsupervised Representation Learning for Time Series with Temporal Neighborhood Coding

**Sana Tonekaboni***
University of Toronto & Vector Institute
The Hospital for Sick Children
`stonekaboni@cs.toronto.edu`

**Danny Eytan**
The Hospital for Sick Children
`biliary.colic@gmail.com`

**Anna Goldengerg**
University of Toronto & Vector Institute
The Hospital for Sick Children
`anna.goldenberg@utoronto.ca`

## Abstract

Time series are often complex and rich in information but sparsely labeled and therefore challenging to model. In this paper, we propose a self-supervised framework for learning generalizable representations for non-stationary time series. Our approach, called Temporal Neighborhood Coding (TNC), takes advantage of the local smoothness of a signal's generative process to define neighborhoods in time with stationary properties. Using a debiased contrastive objective, our framework learns time series representations by ensuring that in the encoding space, the distribution of signals from within a neighborhood is distinguishable from the distribution of non-neighboring signals. Our motivation stems from the medical field, where the ability to model the dynamic nature of time series data is especially valuable for identifying, tracking, and predicting the underlying patients' latent states in settings where labeling data is practically impossible. We compare our method to recently developed unsupervised representation learning approaches and demonstrate superior performance on clustering and classification tasks for multiple datasets.

## 1 Introduction

Real-world time-series data is high dimensional, complex, and has unique properties that bring about many challenges for data modeling (Yang & Wu, 2006). In addition, these signals are often sparsely labeled, making it even more challenging for supervised learning tasks. Unsupervised representation learning can extract informative low-dimensional representations from raw time series by leveraging the data's inherent structure, without the need for explicit supervision. These representations are more generalizable and robust, as they are less specialized for solving a single supervised task. Unsupervised representation learning is well studied in domains such as vision (Donahue & Simonyan, 2019; Denton et al., 2017; Radford et al., 2015) and natural language processing (Radford et al., 2017; Young et al., 2018; Mikolov et al., 2013), but has been underexplored in the literature for time series settings. Frameworks designed for time series need to be efficient and scalable because signals encountered in practice can be long, high dimensional, and high frequency. Moreover, it should account for and be able to model dynamic changes that occur within samples, i.e., non-stationarity of signals.

The ability to model the dynamic nature of time series data is especially valuable in medicine. Health care data is often organized as a time series, with multiple data types, collected from various sources at different sampling frequencies, and riddled with artifacts and missing values. Throughout their stay at the hospital or within the disease progression period, patients transition gradually between distinct

---

*`http://www.cs.toronto.edu/~stonekaboni/`

clinical states, with periods of relative stability, improvement, or unexpected deterioration, requiring escalation of care that alters the patient's trajectory. A particular challenge in medical time-series data is the lack of well-defined or available labels that are needed for identifying the underlying clinical state of an individual or for training models aimed at extracting low-dimensional representations of these states. For instance, in the context of critical-care, a patient's stay in the critical care unit (CCU) is captured continuously via streaming physiological signals by the bedside monitor. Obtaining labels for the patient's state for extended periods of these signals is practically impossible as the underlying physiological state can be unknown even to the clinicians. This further motivates the use of unsupervised representation learning in these contexts. Learning rich representations can be crucial in facilitating the tracking of disease progression, predicting the future trajectories of the patients, and tailoring treatments to these underlying states.

In this paper, we propose a self-supervised framework for learning representations for complex multivariate non-stationary time series. This approach, called Temporal Neighborhood Coding (TNC), is designed for temporal settings where the latent distribution of the signals changes over time, and it aims to capture the progression of the underlying temporal dynamics. TNC is efficient, easily scalable to high dimensions, and can be used in different time series settings. We assess the quality of the learned representations on multiple datasets and show that the representations are general and transferable to many downstream tasks such as classification and clustering. We further demonstrate that our method outperforms existing approaches for unsupervised representation learning, and it even performs closely to supervised techniques in classification tasks. The contributions of this work are three-fold:

1. We present a novel neighborhood-based unsupervised learning framework for *non-stationary* multivariate time series data.

2. We introduce the concept of a temporal neighborhood with stationary properties as the distribution of similar windows in time. The neighborhood boundaries are determined automatically using the properties of the signal and statistical testing.

3. We incorporate concepts from Positive Unlabeled Learning, specifically, sample weight adjustment, to account for potential bias introduced in sampling negative examples for the contrastive loss.

## 2 METHOD

We introduce a framework for learning representations that encode the underlying state of a multivariate, non-stationary time series. Our self-supervised approach, TNC, takes advantage of the local smoothness of the generative process of signals to learn generalizable representations for windows of time series. This is done by ensuring that in the representation space, the distribution of signals proximal in time is distinguishable from the distribution of signals far away, i.e., proximity in time is identifiable in the encoding space. We represent our multivariate time series signals as $X \in R^{D \times T}$, where $D$ is the number of features and $T$ is the number of measurements over time. $X_{[t-\frac{\delta}{2}, t+\frac{\delta}{2}]}$ represents a window of time series of length $\delta$, centered around time $t$, that includes measurements of all features taken in the interval $[t - \frac{\delta}{2}, t + \frac{\delta}{2}]$. Throughout the paper, we refer to this window as $W_t$ for notational simplicity. Our goal is to learn the underlying representation of $W_t$, and by sliding this window over time, we can obtain the trajectory of the underlying states of the signal.

We define the temporal neighborhood ($N_t$) of a window $W_t$ as the set of all windows with centroids $t^*$, sampled from a normal distribution $t^* \sim \mathcal{N}(t, \eta \cdot \delta)$. Where $\mathcal{N}$ is a Gaussian centered at $t$, $\delta$ is the size of the window, and $\eta$ is the parameter that defines the range of the neighborhood. Relying on the local smoothness of a signal's generative process, the neighborhood distribution is characterized as a Gaussian to model the gradual transition in temporal data, and intuitively, it approximates the distribution of samples that are similar to $W_t$. The $\eta$ parameter determines the neighborhood range and depends on the signal characteristics and how gradual the time series's statistical properties change over time. This can be set by domain experts based on prior knowledge of the signal behavior, or for more robust estimation, it can be determined by analyzing the stationarity properties of the signal for every $W_t$. Since the neighborhood represents similar samples, the range should identify the approximate time span within which the signal remains stationary, and the generative process does not change. For this purpose, we use the Augmented Dickey-Fuller (ADF) statistical test to

determine this region for every window. Proper estimation of the neighborhood range is an integral part of the TNC framework. If $\eta$ is too small, many samples from within a neighborhood will overlap, and therefore the encoder would only learn to encode the overlapping information. On the other hand, if $\eta$ is too big, the neighborhood would span over multiple underlying states, and therefore the encoder would fail to distinguish the variation among these states. Using the ADF test, we can automatically adjust the neighborhood for every window based on the signal behavior. More details on this test and how it is used to estimate $\eta$ is described in section 2.

Now, assuming windows within a neighborhood possess similar properties, signals outside of this neighborhood, denoted as $\bar{N}_t$, are considered non-neighboring windows. Samples from $\bar{N}_t$ are likely to be different from $W_t$, and can be considered as negative samples in a context of a contrastive learning framework. However, this assumption can suffer from the problem of *sampling bias*, common in most contrastive learning approaches (Chuang et al., 2020; Saunshi et al., 2019). This bias occurs because randomly drawing negative examples from the data distribution may result in negative samples that are actually similar to the reference. This can significantly impact the learning framework's performance, but little work has been done on addressing this issue (Chuang et al., 2020). In our context, this can happen when there are windows from $\bar{N}_t$ that are far away from $W_t$, but have the same underlying state. To alleviate this bias in the TNC framework, we consider samples from $\bar{N}_t$ as unlabeled samples, as opposed to negative, and use ideas from Positive-Unlabeled (PU) learning to accurately measure the loss function. In reality, even though samples within a neighborhood are all similar, we cannot make the assumption that samples outside this region are necessarily different. For instance, in the presence of long term seasonalities, signals can exhibit similar properties at distant times. In a healthcare context, this can be like a stable patient that undergoes a critical condition, but returns back to a stable state afterwards.

In PU learning, a classifier is learned using labeled data drawn from the positive class ($P$) and unlabeled data ($U$) that is a mixture of positive and negative samples with a positive class prior $\pi$ (Du Plessis et al., 2014; Kiryo et al., 2017; Du Plessis & Sugiyama, 2014). Existing PU learning methods fall under two categories based on how they handle the unlabeled data: 1) methods that identify negative samples from the unlabeled cohort (Li & Liu, 2003); 2) methods that treat the unlabeled data as negative samples with smaller weights (Lee & Liu, 2003; Elkan & Noto, 2008). In the second category, unlabeled samples should be properly weighted in the loss term in order to train an unbiased classifier. Elkan & Noto (2008) introduces a simple and efficient way of approximating the expectation of a loss function by assigning individual weights $w$ to training examples from the unlabeled cohort. This means each sample from the neighborhood is treated as a positive example with unit weight, while each sample from $\bar{N}$ is treated as a combination of a positive example with weight $w$ and a negative example with complementary weight $1 - w$. In the original paper (Elkan & Noto, 2008), the weight is defined as the probability for a sample from the unlabeled set to be a positive sample, i.e. $w = p(y = 1|x)$ for $x \in U$. In the TNC framework, this weight represents the probability of having samples similar to $W_t$ in $\bar{N}$. By incorporating weight adjustment into the TNC loss (Equation 1), we account for possible positive samples that occur in the non-neighboring distribution. $w$ can be approximated using prior knowledge of the underlying state distribution or tuned as a hyperparameter. Appendix A.6 explains how the weight parameter is selected for our different experiment setups and also demonstrates the impact of weight adjustment on performance for downstream tasks.

After defining the neighborhood distribution, we train an objective function that encourages a distinction between the representation of samples of the same neighborhood from the outside samples. An ideal encoder preserves the neighborhood properties in the encoding space. Therefore representations $Z_l = Enc(W_l)$ of samples from a neighborhood $W_l \in N_t$, can be distinguished from representation $Z_k = Enc(W_k)$ of samples from outside the neighborhood $W_k \in \bar{N}_t$. TNC is composed of two main components:

1. An Encoder $Enc(W_t)$ that maps $W_t \in \mathbb{R}^{D \times \delta}$ to a representation $Z_t \in \mathbb{R}^M$, in a lower dimensional space ($M \ll D \times \delta$), where $D \times \delta$ is the total number of measurements in $W_t$.

2. A Discriminator $\mathcal{D}(Z_t, Z)$ that approximates the probability of $Z$ being the representation of a window in $N_t$. More specifically, it receives two samples from the encoding space and predicts the probability of those samples belonging to the same temporal neighborhood.

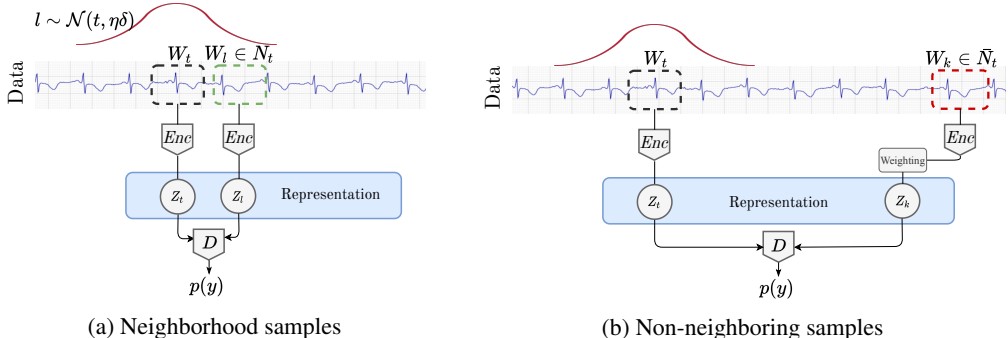

(a) Neighborhood samples          (b) Non-neighboring samples

Figure 1: Overview of the TNC framework components. For each sample window $W_t$ (indicated with the dashed black box), we first define the neighborhood distribution. The encoder learns the distribution of windows sampled from $N_t$ and $\bar{N}_t$, in the representation space. Then samples from these distributions are fed into the discriminator alongside $Z_t$, to predict the probability of the windows being in the same neighborhood.

TNC is a general framework; therefore, it is agnostic to the nature of the time series and the architecture of the encoder. The encoder can be any parametric model that is well-suited to the signal properties (Oord et al., 2016; Bai et al., 2018; Fawaz et al., 2019). For the Discriminator $\mathcal{D}(Z_t, Z)$ we use a simple multi-headed binary classifier that outputs 1 if $Z$ and $Z_t$ are representations of neighbors in time, and 0 otherwise. In the experiment section, we describe the architectural details of the models used for our experiments in more depth.

Figure 1 provides a summary overview of the TNC framework. We formalize the objective function of our unsupervised learning framework in Equation 1. In essence, we would like the probability likelihood estimation of the Discriminator to be accurate, i.e., close to 1 for the representation of neighboring samples and close to 0 for windows far apart. Samples from the non-neighboring region ($\bar{N}$) are weight-adjusted using the $w$ parameters to account for positive samples in this distribution.

$$\mathcal{L} = -\mathbb{E}_{W_t \sim X}[\mathbb{E}_{W_l \sim N_t}[\log \underbrace{\mathcal{D}(Z_t, Z_l)}_{\mathcal{D}(Enc(W_t), Enc(W_l))}] + \mathbb{E}_{W_k \sim \bar{N}_t}[(1-w_t) \times \log{(1 - \underbrace{\mathcal{D}(Z_t, Z_k)}_{\mathcal{D}(Enc(W_t), Enc(W_k))})} + w_t \times \log \mathcal{D}(Z_t, Z_k)]]$$

(1)

We train the encoder and the discriminator hand in hand by optimizing for this objective. Note that the Discriminator is only part of training and will not be used during inference. Similar to the encoder, it can be approximated using any parametric model. However, the more complex the Discriminator, the harder it becomes to interpret the latent space's decision boundaries since it allows similarities to be mapped on complex nonlinear relationships.

**Defining the neighborhood parameter using the ADF test:** As mentioned earlier, the neighborhood range can be specified using the characteristics of the data. In non-stationary time series, the generative process of the signals changes over time. We define the temporal neighborhood around every window as the region where the signal is relatively stationary. Since a signal may remain in an underlying state for an unknown amount of time, each window's neighborhood range may vary in size and must be adjusted to signal behavior. To that end, we use the Augmented Dickey-Fuller (ADF) statistical test to derive the neighborhood range $\eta$. The ADF test belongs to a category of tests called "Unit Root Test", and is a method for testing the stationarity of a time series. For every $W_t$, we want to find the neighborhood range around that window that indicates a stationary region. To determine this, we start from $\eta = 1$ and gradually increase the neighborhood size $\eta$, measuring the $p$-value from the test at every step. Once $p$-value is above a threshold (in our setting $0.01$), it means that it fails to reject the null hypothesis and suggests that within this neighborhood region, the signal is no longer stationary. This way, we find the widest neighborhood within which the signal remains relatively stationary. Note that the window size $\delta$ is constant throughout the experiment, and during ADF testing, we only adjust the neighborhood's width.

## 3    EXPERIMENTS

We evaluate our framework's usability on multiple time series datasets with dynamic latent states that change over time. We compare classification performance and clusterability against two state-of-the-art approaches for unsupervised representation learning for time series: 1. Contrastive Predictive Coding (CPC) (Oord et al., 2018) that uses predictive coding principles to train the encoder on a probabilistic contrastive loss. 2. Triplet-Loss (T-Loss), introduced in (Franceschi et al., 2019), which employs time-based negative sampling and a triplet loss to learn representations for time series windows. The triplet loss objective ensures similar time series have similar representations by minimizing the pairwise distance between positive samples (subseries) while maximizing it for negative ones. (See Appendix A.2 for more details on each baseline.)

For a fair comparison and to ensure that the difference in performance is not due to the differences in the models' architecture, the same encoder network is used across all compared baselines. Our objective is to compare the performance of the learning frameworks, agnostic to the encoder's choice. Therefore, we selected simple architectures to evaluate how each framework can use a simple encoder's limited capacity to learn meaningful representations. We assess the generalizability of the representations by 1) evaluating clusterability in the encoding space and 2) using the representations for a downstream classification task. In addition to the baselines mentioned above, we also compare clusterability performance with unsupervised K-means and classification with a K-Nearest Neighbor classifier, using Dynamic Time Warping (DTW) to measure time series distance. All models are implemented using Pytorch 1.3.1 and trained on a machine with Quadro 400 GPU [1]. Below we describe the datasets for our experiments in more detail.

### 3.1    SIMULATED DATA

The simulated dataset is designed to replicate very long, non-stationary, and high-frequency time series for which the underlying dynamics change over time. Our generated time series consists of 2000 measurements for 3 features, generated from 4 different underlying states. We use a Hidden Markov Model (HMM) to generate the random latent states over time, and in each state, the time series is generated from a different generative process, including Gaussian Processes (GPs) with different kernel functions and Nonlinear Auto-regressive Moving Average models with different sets of parameters ($\alpha$ and $\beta$). Besides, for it to further resemble realistic (e.g., clinical) time series, two features are always correlated. More details about this dataset are provided in the Appendix A.1. For this experimental setup, we use a two-directional, single-layer recurrent neural network encoder. We have selected this simple architecture because it handles time series with variable lengths, and it easily extends to higher-dimensional inputs. The encoder model encodes multi-dimensional signal windows of $\delta = 50$ into 10 dimensional representation vectors. The window size is selected such that it is long enough to contain information about the underlying state but not too long to span over multiple underlying states. A more detailed discussion on the window size choice is presented in Appendix A.4.

### 3.2    CLINICAL WAVEFORM DATA

For a real-world clinical experiment, we use the MIT-BIH Atrial Fibrillation dataset (Moody, 1983). This dataset includes 25 long-term Electrocardiogram (ECG) recordings (10 hours in duration) of human subjects with atrial fibrillation. It consists of two ECG signals; each sampled at 250 Hz. The signals are annotated over time for the following different rhythm types: 1) Atrial fibrillation, 2) Atrial flutter, 3) AV junctional rhythm, and 4) all other rhythms. Our goal in this experiment is to identify the underlying type of arrhythmia for each sample without any information about the labels. This dataset is particularly interesting and makes this experiment challenging due to the following special properties:

- The underlying heart rhythm changes over time in each sample. This is an opportunity to evaluate how different representation learning frameworks can handle alternating classes in non-stationary settings;

---

[1]Code    implementation    can    be    found    at    https://github.com/sanatonek/TNC_
representation_learning

- The dataset is highly imbalanced, with atrial flutter and AV junctional rhythm being present in fewer than $0.1\%$ of the measurements. Data imbalance poses many challenges for downstream classification, further motivating the use of unsupervised representation learning;

- The dataset has samples from a small number of individuals, but over an extended period (around 5 million data points). This realistic scenario, common in healthcare data, shows that our framework is still powerful in settings with a limited number of samples.

The simple RNN encoder architecture used for other experiment setups cannot model the high-frequency ECG measurements. Therefore, inspired by state-of-the-art architectures for ECG classification problems, the encoder $Enc$ used in this experiment is a 2-channel, 1-dimensional strided convolutional neural network that runs directly on the ECG waveforms. We use six convolutional layers with a total down-sampling factor of 16. The window size is 2500 samples, meaning that each convolutional filter covers at least half a second of ECG recording, and the representations are summarized in a 64-dimensional vector.

### 3.3 HUMAN ACTIVITY RECOGNITION (HAR) DATA

Human Activity Recognition (HAR) is the problem of predicting the type of activity using temporal data from accelerometer and gyroscope measurements. We use the HAR dataset from the UCI Machine Learning Repository [2] that includes data collected from 30 individuals using a smartwatch. Each person performs six activities: 1) walking, 2) walking upstairs, 3) walking downstairs, 4) sitting, 5) standing, and 6) laying. The time-series measurements are pre-processed to extract 561 features. For our purpose, we concatenate the activity samples from every individual over time using the subject identifier to build the full-time series for each subject, which includes continuous activity change. Similar to the simulated data setting, we use a single-layer RNN encoder. The selected window size is 4, representing about 15 seconds of recording, and the representations are encoded in a 10-dimensional vector space.

## 4 RESULTS

In this section we present the results for clusterability of the latent representations and downstream classification performance for all datasets and across all baselines. Clusterability indicates how well each method recovers appropriate states, and classification assesses how informative our representations are for downstream tasks.

### 4.1 EVALUATION: CLUSTERABILITY

Many real-world time series data have underlying multi-category structure, naturally leading to representations with clustering properties. Encoding such general priors is a property of a good representation (Bengio et al., 2013). In this section, we assess the distribution of the representations in the encoding space. If information of the latent state is properly learned and encoded by the framework, the representations of signals from the same underlying state should cluster together. Figures 2a, 2b, and 2c show an example of this distribution for simulated data across compared approaches. Each plot is a 2-dimensional t-SNE visualization of the representations where each data point in the scatter plot is an encoding $Z \in R^{10}$ that represents a window of size $\delta = 50$ of a simulated time series. We can see that without any information about the hidden states, representations learned using TNC cluster windows from the same hidden state better than the alternative approaches. The results show that CPC and Triplet Loss have difficulty separating time series that are generated from non-linear auto-regressive moving average (NARMA) models with variable regression parameters.

To compare the representation clusters' consistency for each baseline, we use two very common cluster validity indices, namely, the Silhouette score and the Davies-Bouldin index. We use K-means clustering in the representation space to measure these clusterability scores. The Silhouette score measures the similarity of each sample to its own cluster, compared to other clusters. The values can range from $-1$ to $+1$, and a greater score implies a better cohesion. The Davies-Bouldin Index

---

[2]https://archive.ics.uci.edu/ml/datasets/human+activity+recognition+using+smartphones

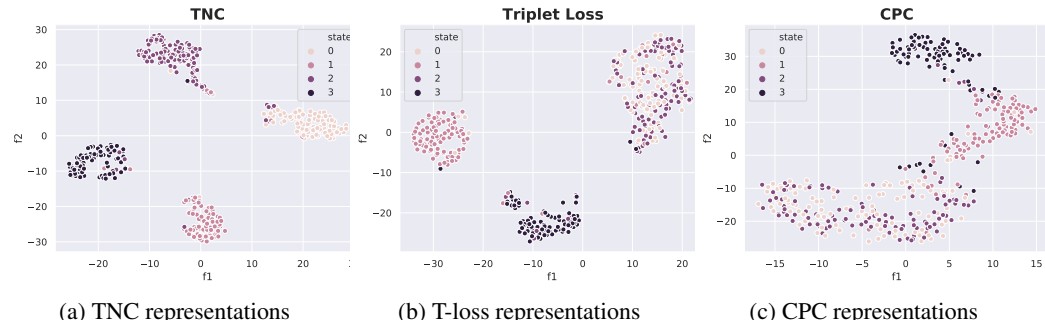

(a) TNC representations  (b) T-loss representations  (c) CPC representations

Figure 2: T-SNE visualization of signal representations for the simulated dataset across all baselines. Each data point in the plot presents a 10-dimensional representation of a window of time series of size $\delta = 50$, and the color indicates the latent state of the signal window. See Appendix A.7 for similar plots from different datasets.

measures intra-cluster similarity and inter-cluster differences. This is a positive index score, where smaller values indicate low within-cluster scatter and large separation between clusters. Therefore, a lower score represents better clusterability (more details on the cluster validity scores and how they are calculated can be found in Appendix A.5).

| | Simulation | | ECG Waveform | | HAR | |
|---|---|---|---|---|---|---|
| Method | Silhouette ↑ | DBI ↓ | Silhouette ↑ | DBI ↓ | Silhouette ↑ | DBI ↓ |
| **TNC** | **0.71±0.01** | **0.36±0.01** | **0.44±0.02** | **0.74±0.04** | **0.61±0.02** | **0.52±0.04** |
| CPC | 0.51±0.03 | 0.84±0.06 | 0.26±0.02 | 1.44±0.04 | 0.58±0.02 | 0.57±0.05 |
| T-Loss | 0.61±0.08 | 0.64±0.12 | 0.25±0.01 | 1.30±0.03 | 0.17±0.01 | 1.76±0.20 |
| K-means | 0.01±0.019 | 7.23±0.14 | 0.19±0.11 | 3.65±0.48 | 0.12±0.40 | 2.66±0.05 |

Table 1: Clustering quality of representations in the encoding space for multiple datasets.

Table 1 summarizes the scores for all baselines and across all datasets, demonstrating that TNC is superior in learning representations that can distinguish the latent dynamics of time series. CPC performs closely to Triplet loss on waveform data but performs poorly on the simulated dataset, where signals are highly non-stationary, and transitions are less predictable. However, for the HAR dataset, CPC clusters the states very well because most activities are recorded in a specific order, empowering predictive coding. Triplet loss performs reasonably well in the simulated setting; however, it fails to distinguish states 0 and 2, where signals come from autoregressive models with different parameters and have a relatively similar generative process. Performing K-means on the original time series generally does not generate coherent clusters, as demonstrated by the scores. However, the performance is slightly better in time series like the ECG waveforms, where the signals are formed by consistent shapelets, and therefore the DTW measures similarity more accurately.

## 4.2 EVALUATION: CLASSIFICATION

We further evaluate the quality of the encodings using a classification task. We train a linear classifier to evaluate how well the representations can be used to classify hidden states. The performance of all baselines is compared to a supervised classifier, composed of an encoder and a classifier with identical architectures to that of the unsupervised models, and a K-nearest neighbor classifier that uses DTW metric. The performance is reported as the prediction accuracy and the area under the precision-recall curve (AUPRC) score since AUPRC is a more accurate reflection of model performance for imbalance classification settings like the waveform dataset.

Table 2 demonstrates the classification performance for all datasets. The performance of the classifiers that use TNC representations are closer to the end-to-end supervised model in comparison to CPC and Triplet Loss. This provides further evidence that our encodings capture informative parts of the time series and are generalizable to be used for downstream tasks. In datasets like the HAR, where

|  | Simulation | | ECG Waveform | | HAR | |
|---|---|---|---|---|---|---|
| Method | AUPRC | Accuracy | AUPRC | Accuracy | AUPRC | Accuracy |
| **TNC** | **0.99±0.00** | **97.52±0.13** | **0.55±0.01** | **77.79±0.84** | **0.94±0.007** | **88.32±0.12** |
| CPC | 0.69±0.06 | 70.26±6.48 | 0.42±0.01 | 68.64±0.49 | 0.93±0.006 | 86.43±1.41 |
| T-Loss | 0.78±0.01 | 76.66±1.40 | 0.47±0.00 | 75.51±1.26 | 0.71±0.007 | 63.60±3.37 |
| KNN | 0.42±0.00 | 55.53±0.65 | 0.38±0.06 | 54.76±5.46 | 0.75±0.01 | 84.85±0.84 |
| Supervised | **0.99±0.00** | **98.56±0.13** | **0.67±0.01** | **94.81±0.28** | **0.98±0.00** | **92.03±2.48** |

Table 2: Performance of all baselines in classifying the underlying hidden states of the time series, measured as the accuracy and AUPRC score.

an inherent ordering usually exists in the time series, CPC performs reasonably. However, in datasets with increased non-stationarity, the performance drops. Triplet Loss is also a powerful framework, but since it samples positive examples from overlapping windows of time series, it is vulnerable to map the overlaps into the encoding and, therefore, fail to learn more general representations. TNC, on the other hand, samples similar windows from a wider distribution, defined by the temporal neighborhood, where many of the neighboring signals do not necessarily overlap. The lower performance of the CPC and Triplet Loss methods can also be partly because none of these methods explicitly account for the potential sampling bias that happens when randomly selected negative examples are similar to the reference $W_t$.

## 4.3 EVALUATION: TRAJECTORY

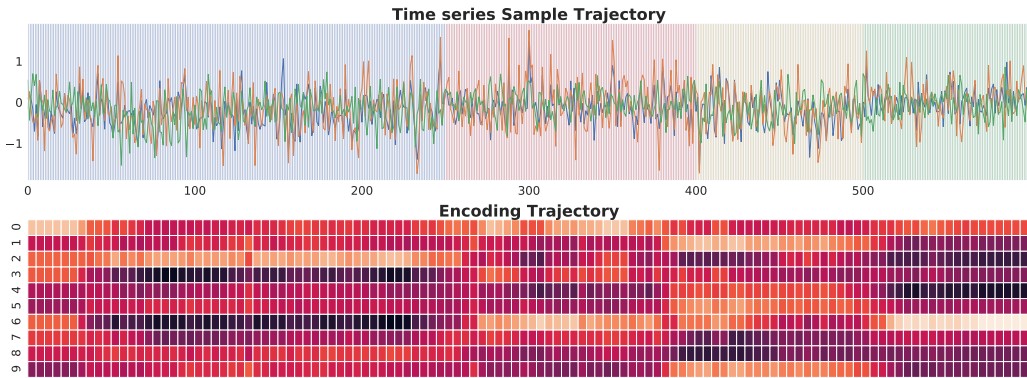

Figure 3: Trajectory of a signal encoding from the simulated dataset. The top plot shows the original time series with shaded regions indicating the underlying state. The bottom plot shows the 10 dimensional encoding of the sliding windows $W_t$ where $\delta = 50$.

This section investigates the trajectories of our learned encodings over time to understand how the state transitions are captured and modeled in the representation space. This is an important property for non-stationary time series where underlying states change over time, and capturing those changes is critical in many application domains such as healthcare. Figure 3 shows a sample from the simulated dataset. The top panel shows the signal measurements over time, and the shaded regions indicate the underlying latent states. The bottom panel illustrates the 10-dimensional representation of a sliding window $W_t$ estimated over time. From the bottom panel of Figure 3, we can see that the encoding pattern changes at state transitions and settle into a different pattern, corresponding to the new state. This change happens at every transition, and we can see the distinct patterns for all 4 underlying states in the representations. This analysis of the trajectory of change could be very informative for the users' post-analysis; for instance, in clinical applications, it could help clinicians visualize the evolution of the patient state over time and plan treatment based on the state progression

## 5 RELATED WORK

While integral for many applications, unsupervised representation learning has been far less studied for time series (Längkvist et al., 2014), compared to other domains such as vision or natural language processing (Denton et al., 2017; Radford et al., 2015; Gutmann & Hyvärinen, 2012; Wang & Gupta, 2015). One of the earliest approaches to unsupervised end-to-end representation learning in time series is the use of auto-encoders (Choi et al., 2016a; Amiriparian et al., 2017; Malhotra et al., 2017) and seq-to-seq models (Lyu et al., 2018), with the objective to train an encoder jointly with a decoder that reconstructs the input signal from its learned representation. Using fully generative models like variational auto-encoders is also useful for imposing properties like disentanglement, which help with the interpretability of the representations (Dezfouli et al., 2019). However, in many cases, like for high-frequency physiological signals, the reconstruction of complex time series can be challenging; therefore, more novel approaches are designed to avoid this step. Contrastive Predictive Coding (Oord et al., 2018; Löwe et al., 2019) learns representations by predicting the future in latent space, eliminating the need to reconstruct the full input. The representations are such that the mutual information between the original signal and the concept vector is maximally preserve using a lower bound approximation and a contrastive loss. Very similarly, in Time Contrastive Learning (Hyvarinen & Morioka, 2016), a contrastive loss is used to predict the segment-ID of multivariate time-series as a way to extract representation. Franceschi et al. (2019) employs time-based negative sampling and a triplet loss to learn scalable representations for multivariate time series. Some other approaches use inherent similarities in temporal data to learn representations without supervision. For instance, in similarity-preserving representation learning (Lei et al., 2019), learned encodings are constrained to preserve the pairwise similarities that exist in the time domain, measured by DTW distance. Another group of approaches combines reconstruction loss with clustering objectives to cluster similar temporal patterns in the encoding space (Ma et al., 2019).

In healthcare, learning representation of rich temporal medical data is extremely important for understanding patients' underlying health conditions. However, most of the existing approaches for learning representations are designed for specific downstream tasks and require labeling by experts (Choi et al., 2016b;c; Tonekaboni et al., 2020). Examples of similar works to representation learning in the field of clinical ML include computational phenotyping for discovering subgroups of patients with similar underlying disease mechanisms from temporal clinical data (Lasko et al., 2013; Suresh et al., 2018; Schulam et al., 2015), and disease progression modeling, for learning the hidden vector of comorbidities representing a disease over time (Wang et al., 2014; Alaa & van der Schaar, 2019).

## 6 CONCLUSION

This paper presents a novel unsupervised representation learning framework for complex multivariate time series, called Temporal Neighborhood Coding (TNC). This framework is designed to learn the underlying dynamics of non-stationary signals and to model the progression over time by defining a temporal neighborhood. The problem is motivated by the medical field, where patients transition between distinct clinical states over time, and obtaining labels to define these underlying states is challenging. We evaluate the performance of TNC on multiple datasets and show that our representations are generalizable and can easily be used for diverse tasks such as classification and clustering. We finally note that TNC is flexible to be used with arbitrary encoder architectures; therefore, the framework is applicable to many time series data domains. Moreover, in addition to tasks presented in this paper, general representations can be used for several other downstream tasks, such as anomaly detection, which is challenging in supervised learning settings for time series data in sparsely labeled contexts.

### ACKNOWLEDGMENTS

Resources used in preparing this research were provided, in part, by the Government of Canada through CIFAR, and companies sponsoring the Vector Institute. This research was undertaken, in part, thanks to funding from the Canadian Institute of Health Research (CIHR) and the Natural Sciences and Engineering Research Council of Canada (NSERC).

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

## A    APPENDIX

### A.1    SIMULATED DATASET

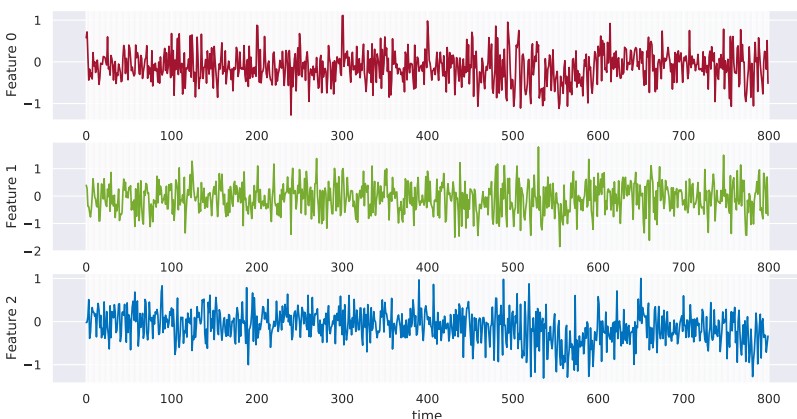

Figure A.1: A normalized time series sample from the simulated dataset. Each row represents a single feature, and the shaded regions indicate one of the 4 underllying simulated states.

The simulated time series consists of 3 features generated from different underlying hidden states. Figure A.1 shows a sample from this dataset. Each panel in the figure shows one of the features, and the shaded regions indicate the underlying state of the signal in that period. We use a Hidden Markov Model (HMM) to generate these random latent states over time. The transition probability is set equal to %5 for switching to an alternating state, and %85 for not changing state. In each state, the time series is generated from a different signal distribution. Table 3 describes the generative process of each signal feature in each state. Note that feature 1 and 2 are always correlated, mainly to mimic realistic clinical time series. As an example, physiological measurements like pulse rate and heart rate are always correlated.

|           | State 1           | State 2          | State 3           | State 4          |
|-----------|-------------------|------------------|-------------------|------------------|
| Feature 1 | GP (periodic)     | NARMA$_\alpha$   | GP (Squared Exp.) | NARMA$_\beta$    |
| Feature 2 | GP (periodic)     | NARMA$_\alpha$   | GP (Squared Exp.) | NARMA$_\beta$    |
| Feature 3 | GP (Squared Exp.) | NARMA$_\beta$    | GP (periodic)     | NARMA$_\alpha$   |

Table 3: Signal distributions for each time series feature of the simulated dataset

In the state 1, the correlated features are generated by a Gaussian Process (GP) with a periodic kernel. Feature 3, which is uncorrelated with the other two features, comes from another GP with a squared exponential kernel. In addition to GPs, we also have multiple Non-Linear Auto-Regressive Moving Average (NARMA) time series models. The linear function of NARMA$_\alpha$ and NARMA$_\beta$ are shown in Equation 2 and 3.

$$\text{NARMA}_\alpha : y(k+1) = 0.3y(k) + 0.05y(k)\sum_{i=0}^{n-1} y(k-i) + 1.5u(k-(n-1))u(k) + 0.1 \quad (2)$$

$$\text{NARMA}_\beta : y(k+1) = 0.1y(k) + 0.25y(k)\sum_{i=0}^{n-1} y(k-i) + 2.5u(k-(n-1))u(k) - 0.005 \quad (3)$$

A white Gaussian noise with $\sigma = 0.3$ is added to all signals, and overall, the dataset consists of 500 instances of $T = 2000$ measurements.

## A.2 BASELINE IMPLEMENTATION DETAILS

Implementation of all baselines are included in the code base for reproducibility purposes, and hyper-parameters for all baselines are tuned using cross-validation.

**Contrastive Predictive Coding (CPC):** The CPC baseline first processes the sequential signal windows using an encoder $Z_t = Enc(X_t)$, with a similar architecture to the encoders of other baselines. Next, an autoregressive model $g_{ar}$ aggregates all the information in $Z_{\leq t}$ and summarizes it into a context latent representation $c_t = g_{ar}(Z_{\leq t})$. In our implementation, we have used a single layer, a one-directional recurrent neural network with GRU cell and hidden size equal to the encoding size as the auto-regressor. Like the original paper, the density ratio is estimated using a linear transformation, and the model is trained for 1 step ahead prediction.

**Triplet-Loss (T-Loss):** The triplet loss baseline is implemented using the original code made available by the authors on Github[3].

**KNN and K-means:** These two baselines for classification and clustering are implemented using the tslearn library [4], that integrates distance metrics such as DTW. Note that evaluating DTW is computationally expensive, and the tslearn implementation is not optimized. Therefore, for the waveform data with windows of size 2500, we had to down-sampled the signal frequency by a factor of two.

## A.3 TNC IMPLEMENTATION EXTRA DETAILS

To define the neighborhood range in the TNC framework, as mentioned earlier, we use the Augmented-Dickey Fuller (ADF) statistical test to determine this range ($\eta$) as the region for which the signals remain stationary. More precisely, we gradually increase the range, from a single window size up to 3 times the window size (the upper limit we set), and repeatedly perform the ADF test. We use the $p$-value from this statistical test to determine whether the Null hypothesis can be rejected, meaning that the signal is stationary. At the point where the $p$-value is above our defined threshold (0.01), we can no longer assume that the signal is stationary, and this is where we set the $\eta$ parameter. Now, once the neighborhood is defined, we make sure the non-neighboring samples are taken from the distribution of windows with at least $4 \times \eta$ away from $W_t$, ensuring a low likelihood of belonging to the neighborhood. Note that for implementation of ADF, we use the stats model library [5]. Unfortunately, this implementation is not optimized and does not support GPU computation, therefore evaluating the neighborhood range using ADF slows down TNC framework training. As a future direction, we are working on an optimized implementation of the ADF score for our framework.

## A.4 SELECTING THE WINDOW SIZE

The window size $\delta$ is an important factor in the performance of a representation learning framework, not only for TNC but also for similar baselines such as CPC and triplet loss. Overall, the window size should be selected such that it is long enough to contain information about the underlying state and not too long to span over multiple underlying states. In our settings, we have selected the window sizes based on our prior knowledge of the signals. For instance, in the case of an ECG signal, the selected window size is equivalent to 7 seconds of recording, which is small enough such that the ECG remains in a stable state and yet has enough information to determine that underlying state. Our understanding of the time series data can help us select an appropriate window size, but we can also experiment with different $\delta$ to learn this parameter. Table 4 shows classification performance results for the simulation setups, under different window sizes. We can clearly see the drop in performance for all baseline methods when the window size is too small or too large.

---

[3] https://github.com/White-Link/UnsupervisedScalableRepresentationLearningTimeSeries
[4] https://tslearn.readthedocs.io/en/stable/index.html
[5] https://www.statsmodels.org/dev/generated/statsmodels.tsa.stattools.adfuller.html

| | $\delta = 10$ | | $\delta = 50$ | | $\delta = 100$ | |
|---|---|---|---|---|---|---|
| | AUPRC | Accuracy | AUPRC | Accuracy | AUPRC | Accuracy |
| TNC | $0.74 \pm 0.01$ | $71.60 \pm 0.59$ | $0.99 \pm 0.00$ | $97.52 \pm 0.13$ | $0.84 \pm 0.11$ | $84.25 \pm 9.08$ |
| CPC | $0.49 \pm 0.02$ | $51.85 \pm 1.81$ | $0.69 \pm 0.06$ | $70.26 \pm 6.48$ | $0.49 \pm 0.05$ | $56.65 \pm 0.81$ |
| T-Loss | $0.48 \pm 0.06$ | $56.70 \pm 1.07$ | $0.78 \pm 0.01$ | $76.66 \pm 1.14$ | $0.73 \pm 0.008$ | $73.29 \pm 1.58$ |

Table 4: Downstream classification performance for different window size $\delta$ on the simulated dataset

## A.5 CLUSTERING METRICS

Most cluster validity measures assess certain structural properties of a clustering result. In our evaluation, we have used two measures, namely the Silhouette score and Davies-Bouldin index, to evaluate the representations' clustering quality. Davies-Bouldin measures intra-cluster similarity (coherence) and inter-cluster differences (separation). Let $\mathcal{C} = \{\mathcal{C}_1, ..., \mathcal{C}_k\}$ be a clustering of a set $D$ of objects. The Davies-Bouldin score is evaluated as follows:

$$DB = \frac{1}{k} \sum_{i=1}^{k} max_j \frac{s(\mathcal{C}) + s(\mathcal{C})}{\delta \mathcal{C}_i \mathcal{C}_j} \qquad (4)$$

Where $s(\mathcal{C})$ measures the scatter within a cluster, and $\delta$ is a cluster to cluster distance measure. On the other hand, the silhouette score measures how similar an object is to its cluster *compared* to other clusters. Both measures are commonly used for the evaluation of clustering algorithms. A comparison of 2 metrics has shown that the Silhouette index produces slightly more accurate results in some cases. However, the Davies-Bouldin index is generally much less complex to compute Petrovic (2006).

## A.6 SETTING THE WEIGHTS FOR PU LEARNING

As mentioned in the Experiment section, the weight parameter in the loss is the probability of sampling a positive window from the non-neighboring region. One way to set this parameter is using prior knowledge of the number and the distribution of underlying states. Another way is to learn it as a hyperparameter. Table 5 shows the TNC loss for different weight parameters. The loss column reports the value measured in Equation 1, and the accuracy shows how well the discriminator identifies the neighboring samples from non-neighboring ones for settings with different weight parameters. To also assess the impact of re-weighting the loss on downstream classification performance, we compared these performance measures for weighted and non-weighted settings. Table 6 demonstrates these results and confirms that weight adjusting the loss for non-neighboring samples improves the quality of learned representations.

| | Simulation | | ECG Waveform | | HAR | |
|---|---|---|---|---|---|---|
| Weight | Loss | Accuracy | Loss | Accuracy | Loss | Accuracy |
| 0.2 | $0.582 \pm 0.002$ | $74.29 \pm 0.61$ | $0.631 \pm 0.011$ | $60.44 \pm 2.56$ | $0.475 \pm 0.004$ | $85.75 \pm 0.5$ |
| 0.1 | $0.571 \pm 0.011$ | $75.41 \pm 0.37$ | $0.637 \pm 0.011$ | $63.67 \pm 1.29$ | $0.413 \pm 0.003$ | $88.21 \pm 1.29$ |
| 0.05 | $0.576 \pm 0.002$ | $75.73 \pm 0.24$ | $0.622 \pm 0.023$ | $66.04 \pm 3.46$ | $0.383 \pm 0.001$ | $87.33 \pm 0.17$ |

Table 5: Training the TNC framework using different weight parameters. The loss is the measured value determined in Equation 1, and the Accuracy is the accuracy of the discriminator.

| Weighting? | Simulation | ECG Waveform | HAR |
|---|---|---|---|
| True | **97.52±0.13** | **77.79±0.84** | **88.32±0.12** |
| False | 97.17±0.44 | 75.26±1.48 | 75.25±13.6 |

Table 6: Downstream classification accuracy on simulated data with the TNC frameworks, using 2 different weighting strategies: 1)Trained with weight adjustment, 2)Trained with $w = 0$.

## A.7 SUPPLEMENTARY FIGURE

### A.7.1 CLINICAL WAVEFORM DATA

In order to understand what TNC framework encodes from the high dimensional ECG signals, we visualize the trajectory of the representations of an individual sample over time. Figure A.3 demonstrates this example, where the top 2 rows are ECG signals from two recording leads and the bottom row demonstrates the representation vectors. We see that around second 40 the pattern in the representations change as a result of an artifact happening in one of the signals. With the help from our clinical expert, we also tried to interpret different patterns in the encoding space. For instance, between time 80 and 130, where features 0-10 become more activated, the heart rate (HR) has increased. Increase in HR can be seen as increased frequency in the ECG signals and is one of the indicators of arrhythmia that we believe TNC has captured. Figure A.2 shows the distribution of the latent encoding of ECG signals for different baselines, with colors indicating the arrhythmia class.

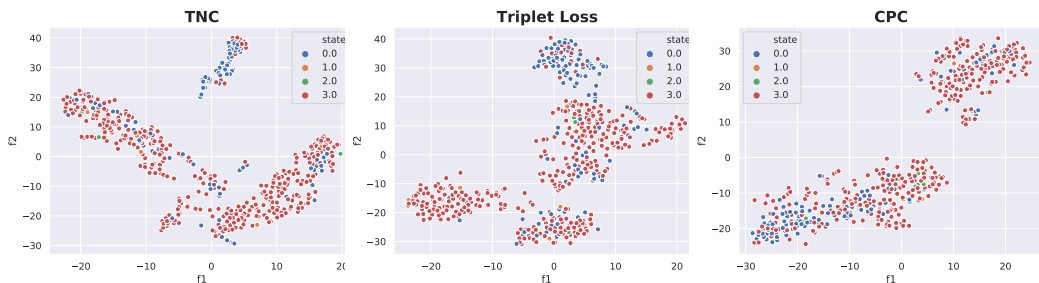

Figure A.2: T-SNE visualization of **waveform** signal representations for unsupervised representation learning baselines. Each point in the plot is a 64 dimensional representation of a window of time series, with the color indicating the latent state.

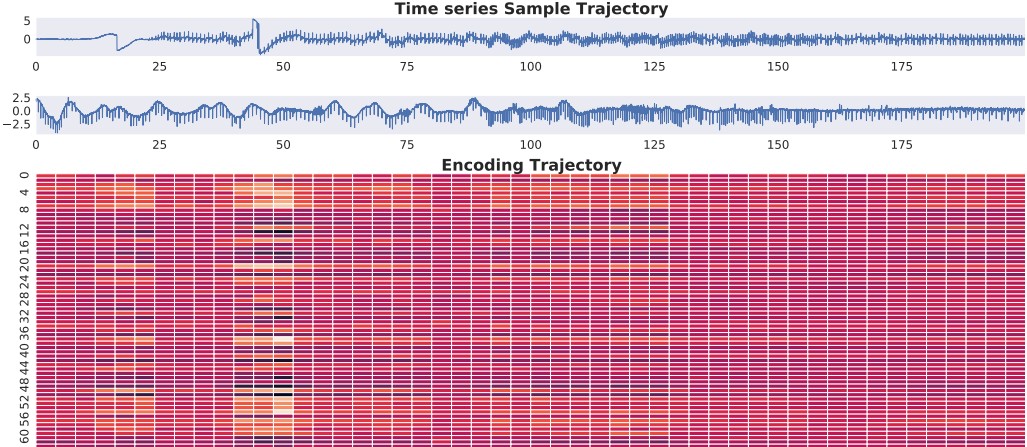

Figure A.3: Trajectory of a **waveform** signal encoding. The top two plots show the ECG recordings from 2 ECG leads. The bottom plot shows the 64 dimensional encoding of the sliding windows $W_t$ where $\delta = 2500$.

### A.7.2 HAR DATA

Figure A.4 and A.5 are similar plots to the ones demonstrated in the previous section, but for the HAR dataset. As shown in Figure A.5, the underlying states of the signal are clearly captured by the TNC framework as different patterns in the latent representations.

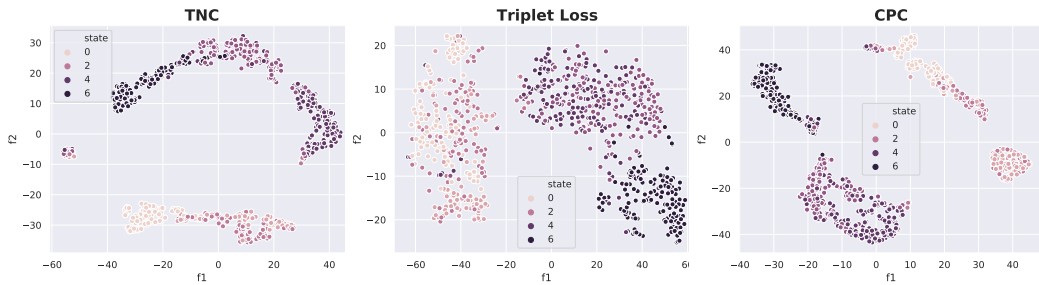

Figure A.4: T-SNE visualization of **HAR** signal representations for all baselines. Each point in the plot is a 10 dimensional representation of a window of $\delta = 4$, with colors indicating latent states.

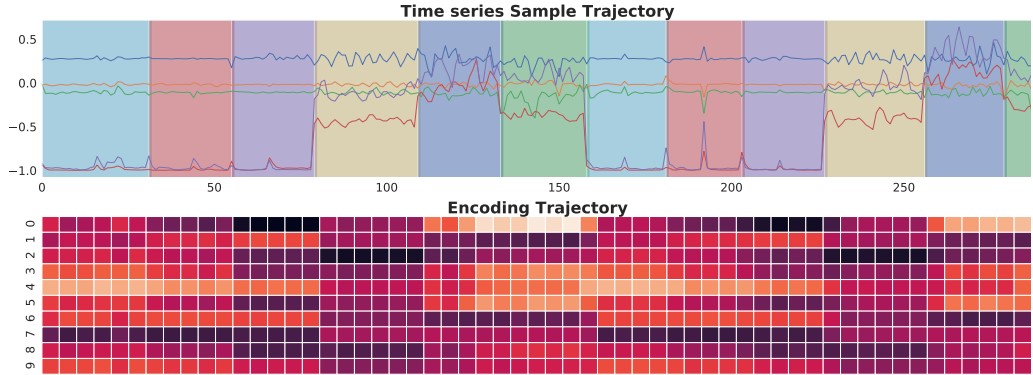

Figure A.5: Trajectory of a **HAR** signal encoding. The top plot shows the original time series with shaded regions indicating the underlying state. The bottom plot shows the 10 dimensional encoding of the sliding windows $W_t$ where $\delta = 4$.

### A.7.3 SIMULATION DATA

In addition to the initial experiment, we also show the trajectory of the encoding for a smaller encoding size (3). In this setting, we have 4 underlying states in the signal, and only 3 dimensions for the encoding.

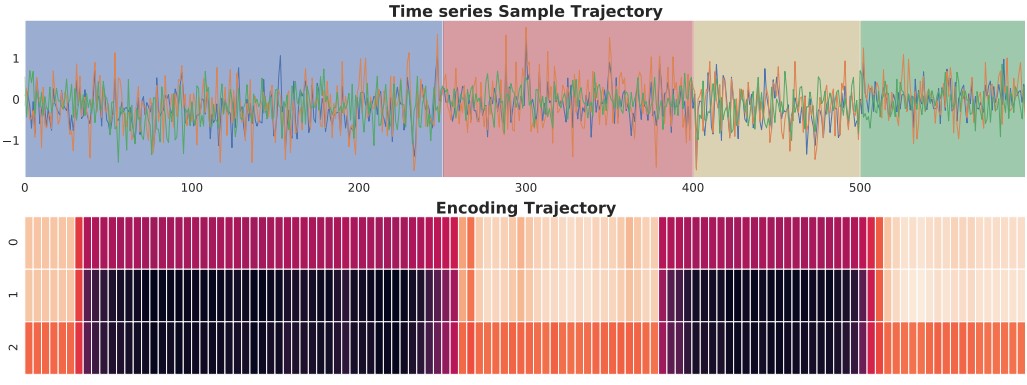

Figure A.6: Trajectory of a **simulation** signal encoding. The top plots shows the signals and the bottom plot shows the 3 dimensional encoding of the sliding windows $W_t$ where $\delta = 50$.

