# OpenReview forum: "Unsupervised Representation Learning for Time Series with Temporal Neighborhood Coding"
_ICLR.cc/2021/Conference — ICLR 2021 Poster_

### Official Review · AnonReviewer2 · 2020-10-26
**Unsupervised Time Series Embeddings for Clustering and Classification**

**Rating:** 8
**Confidence:** 3

**Review:**

The authors propose a novel unsupervised encoding scheme for time series. Utilizing a statistical test for non-stationarity, the authors derive a Temporal Neighborhood Coding (TNC) scheme and combine it with ideas from Positive-Unlabeled (PU) learning to learn informative hidden representations of time series windows. The representations are evaluated in terms of how well they can be clustered and how much they influence classification performance on three data sets. The supreme performance was demonstrated when comparing to the state of the art methods and a $k$NN (for classification) baseline. Furthermore, the authors illustrate how the learnt representations remain interpretable as long as the encoding network is reasonably small.

The presented work is very well motivated and described. The presentation of the ideas is clear and I think the authors did a great job making the manuscript accessible. I like the simplicity of the figures which do not suffer from too little information content. Starting with the analysis of a synthetic data set followed by two real-world data sets is a commendable route to choose for this kind of work. I particularly like the analysis of low sample size-high length ECG data set. I would say job well done! One aspect that is not quite clear to me is how do you combine the ADF test with the fixed-length window sizes that you mention in your experiments?

Some minor comments/questions:

•	I am somewhat confused about the window size information you give in the experiments and for example in Figure 3. Here you say $\delta=50$. I thought the window size is changed dynamically based on the ADF test which leads me to the second question

•	Is the process of determining the window size a preprocessing step or is it part of an end-to-end framework?

•	How do you solve the classification task when you have an encoding for each window? I assume you do not have one specific label for each window. If every single measurement received a ground truth label, isn’t it possible to extract a window annotated with two different labels? Or do you take care that the windows you extract are always centred around the QRS-complex or always starts with the P-wave?

•	It would also be interesting to evaluate how you could perform patient-wise classification (e.g. diagnosis prediction). Maybe it would be an impactful extension to find a good aggregation scheme for the embeddings to summarize single patients.

---

> ### Author Response · Authors · 2020-11-21
> **Thank you for your constructive feedback**
>
> We would like to thank the reviewer for the positive feedback. We are happy to clarify our manuscript in response to the reviewer's questions.
>
> 1.  We would like to clarify that the ADF test intends to find the neighborhood range (governed by $\eta$) and is separate from the window size $\delta$. For every $W_t$, we want to find the neighborhood range around that window that indicates a stationary region. For that purpose, we gradually increase the “neighborhood size $\eta$” and measure the $p$-value using the test. This way we find the widest neighborhood within which the signal remains relatively stationary. We have added more information to the draft to clarify this point.
>
> 2.  The window size is an important design choice across all baselines. Overall, the window size should be selected such that it is long enough to contain information about the underlying state, and not too long to span over multiple underlying states. In our settings, we have selected the window sizes based on our prior knowledge of the signals. For instance, in the case of an ECG signal, the selected window size is equivalent to 7 seconds of recording, which is a window small enough that remains in a stable state, and yet has enough information to determine the state. As also suggested by another reviewer, we have performed experiments to show the impact of window size on thedownstream classification performance (accuracy) of different baselines in the simulated setting. We can see how performance drops for all the methods when the window size is too small or too large. We added this discussion with a full set of resutls to the appendix of our paper.
>
> $\delta$     $\hspace{16pt}$ | $\hspace{24pt}$ 10 $\hspace{24pt}$ |  $\hspace{24pt}$  50 $\hspace{24pt}$   |  $\hspace{24pt}$ 100
>
> ----------------------------------------------------------
> TNC   $\hspace{6pt}$   | $\hspace{6pt}$ 71.60 $\pm$ 0.59 $\hspace{6pt}$ |$\hspace{6pt}$ 97.52 $\pm$ 0.13 $\hspace{6pt}$ | $\hspace{6pt}$ 84.25 $\pm$ 9.08
>
> CPC   $\hspace{6pt}$   | $\hspace{6pt}$  51.85 $\pm$ 1.81 $\hspace{6pt}$  | $\hspace{6pt}$ 70.26 $\pm$ 6.48 $\hspace{6pt}$ | $\hspace{6pt}$ 56.65 $\pm$ 0.81
>
> T-Loss |$\hspace{6pt}$ 56.70 $\pm$ 1.07 $\hspace{6pt}$ | $\hspace{6pt}$ 76.66 $\pm$ 1.14 $\hspace{6pt}$ | $\hspace{6pt}$ 73.29 $\pm$ 1.58
>
>
> 3. For robustness and to ensure time invariance, windows are randomly selected, so for example in the ECG experiment, we do not make all windows centered around the QRS complex.
> Also, to determine the underlying state for each window, we take the majority state, but in general, we want to select the window size to be relatively small such that it does not span over more than one underlying state.
>
> 4. Thank you for bringing up this point, and in fact similar problems (in ICU settings) were our main motivation for this work. Unfortunately, due to the lack of labeled data in such settings, we couldn’t initially evaluate our setup, but we are planning on continuing this work in more complex problem setups such as disease trajectories.

---

### Official Review · AnonReviewer4 · 2020-10-28
**Leveraging the concept of a temporal neighborhood seems beneficial for capturing non-stationary properties of multivariate time series and may have applications in a spectrum of domains (including healthcare). Nevertheless, I feel there are some points (marked as “Improvement points”) which require additional attention. I hope that the authors can address these remarks in the rebuttal period.**

**Rating:** 6
**Confidence:** 4

**Review:**

Paper Summary:

This paper proposes a self-supervised encoder-discriminator based framework for embedding the multivariate time series into a compact fixed dimensional representation. The approach dubbed Temporal Neighborhood Coding (TNC) leverages the concept of a neighborhood in time (with stationary properties), and learns time series representations by ensuring the distribution of neighboring signals is distinguishable from the distribution of non-neighboring signals, in the encoding space. Empirical evidence is provided that such embedding of time series results in clusters of higher quality, as well that use of such obtained representations for supervised tasks outperforms few competitor (unsupervised) approaches.

##########################################################################

Strong points:
1. The neighborhood-based unsupervised learning framework accounts for dynamic changes that occur among samples, i.e. non-stationarity of signals.

2. The introduction of the concept of a temporal neighborhood as the distribution of similar windows in time, whose range can be automatically estimated by analyzing the stationarity of time series with statistical tools such as the Augmented Dickey Fuller (ADF) test.

3. Leveraging the sample weight adjustment, a concept from Positive Unlabeled (PU) learning, to account for the potential bias introduced by sampling negative examples which is a common issue most contrastive approaches suffer from.

4. The proposed TNC outperforms Contrastive Predictive Coding (CPC) and Triplet-Loss (T-Loss), both being state-of-the-art unsupervised representation learning approaches.

5. The paper is well structured and written in a fairly clear and comprehensive manner.

##########################################################################

Improvement points:
1. Efficiency and scalability to high dimensions are mentioned as merits of TNC in the introduction, but are never discussed in further detail. Could the authors elaborate a bit on the components of TNC that allow for its claimed efficiency and scalability and how does TNC compare on these two fronts to other unsupervised representation learning approaches?

2. A bidirectional, single-layer RNN encoder is used in the experiments on the Simulated and HAR datasets, whereas a 2-channel, 1-dimensional strided CNN encoder was applied on the ECG Waveform dataset. The reason behind this decision is not completely clear to me. I would encourage the authors to clearly justify the choice of different encoder architectures in the two cases.

3. In the cases where an RNN architecture is considered for the encoder, the authors could have considered time-aware RNN/LSTM autoencoders as additional baselines. Some of these methods include:
- Baytas, I. M., et al. Patient subtyping via time-aware LSTM networks. In Proceedings of the 23rd ACM SIGKDD (2017).
- Wenjie Pei and David MJ Tax. Unsupervised Learning of Sequence Representations by Autoencoders. arXiv preprint arXiv:1804.00946 (2018).

4. Although the neighborhood range $\eta$ can be automatically determined, choosing the window size $\delta$ still remains unclear, as the performance of TNC under different window sizes was not analysed. Moreover, the selected window sizes seem to be different for different datasets (for instance, a window size of 50 is used for the Simulated data, while a windows size of 4 is used in the case of the HAR dataset). I would like to ask the authors to provide some guidelines on how the window size is (or can be) selected?

5. Some statements appear to be vague and leading away from the main point. This includes usage of phrases such as:
- “Time series are often *complex*” - as in having multiple interacting parts? (I don’t feel that aspect was crucial for paper topic)
- “self-supervised framework for learning *robust* and generalizable representations” - how was the robustness defined and assessed? (no empirical evidence regarding the ‘robustness’ of the learned representations is presented).
- “the representations are general, *transferable*, etc.” - transferable in what sense? (I believe that “applicable across multiple supervised tasks” would be a more appropriate choice of words in this context).

6. There are several minor textual and format typos throughout the paper that can be easily addressed. Some of them are summarized as follows:
- In the second sentence in section 3, consider replacing “state of the art” with “state-of-the-art”.
- In the last sentence of the introductory part of section 3, replace “:” by “.”.
- The term “GPs” is used in section 3.1, but is not defined earlier in the text.
- In section 3.3, “concatenate activity samples” should be replaced by “concatenate the activity samples”.
- In the last couple of sentences in section 4.1, replace “original time series” and “coherent cluster” with “the original time series” and “coherent clusters”, respectively.
- There are certain instances of informal language use, such as “doesn’t” and “don’t”. Please consider replacing them with “does not” and “do not”, respectively.

##########################################################################

Questions during rebuttal period:
Please address the aforementioned remarks/questions.

---

> ### Author Response · Authors · 2020-11-21
> **Thank you for your constructive feedback**
>
> We are greatly indebted to this reviewer for a rigorous and thorough review. Implementing the suggestions has undoubtedly improved the clarity of our paper. Below, we have answered some of the questions mentioned in the feedback.
>
> 1. TNC is a model agnostic framework, therefore with a proper choice of encoder architecture, it can be easily fit to different settings and types of time series. But the most important property of TNC that makes it scalable is that it does not require reconstruction of the input signal. Reconstruction-based approaches are computationally expensive and become harder to train as the signals become more complex with higher dimensions. Also, methods that require measuring the pairwise distance between samples, such as Lei et.al (2017) and derivatives of KNN/K-means, are inefficient and slow during inference. In that sense, contrastive approaches such as TNC, CPC, and Triplet-loss are frameworks that are more efficient and easily scalable to higher-dimensional inputs.
>
> 2. Our goal in the experiments was to compare the performance of different learning frameworks, independent of the encoder architecture, and for that purpose, we used the same encoder model across different methods for each experiment setup. We wanted to see how each framework can use the capacity of a relatively simple encoder to learn meaningful representations, explaining our choice of a simple RNN encoder for the first 2 experiments. For the ECG data, however, a single layer RNN was too simple to model the high-frequency measurements, therefore we had to use a more complex architecture that fits the data better. The architecture that is used is inspired by state-of-the-art architectures for ECG classification problems (A summary of these models is found at https://github.com/hsd1503/DL-ECG-Review). We have added this discussion to the draft as well.
>
> 3. In this paper, we have focused on baselines that do not rely on the reconstruction of the input time series, as these methods can induce large computational costs. However, as suggested by the reviewer, we have performed additional experiments with an RNN autoencoder for learning representations. The architecture of this model is based on Baytas et. al. 2017, without the elapsed time information, since the time series in our experiments are regularly sampled. The table below shows the downstream classification performance of this baseline for simulated and HAR datasets. The performance is comparable to other baselines for HAR dataset, but it deteriorates in the simulation setting, where the signal windows are longer and the time series is noisier. Note that we have not performed extensive hyperparameter tuning for this model and unlike our other baselines, the choice of the encoder architecture is important for an autoencoder setting.
>
> $\hspace{54pt}$ |$\hspace{40pt}$ HAR $\hspace{40pt}$ | $\hspace{40pt}$ Simulation
>
> $\hspace{54pt}$|$\hspace{8pt}$ Accuracy $\hspace{8pt}$| $\hspace{8pt}$ AUPRC$\hspace{8pt}$ |$\hspace{8pt}$ Accuracy$\hspace{8pt}$| $\hspace{8pt}$AUPRC
>
> Auto-encoder | 69.15 $\pm$ 2.06 | 0.62  $\pm$ 0.06 | 43.54  $\pm$ 2.80 | 0.424  $\pm$ 0.030
>
> TNC$\hspace{40pt}$ | 89.41  $\pm$ 0.81 | 0.85  $\pm$ 0.02 | 97.52  $\pm$ 0.13 | 0.99  $\pm$ 0.00

---

> > ### Author Response · Authors · 2020-11-21
> > **Response part 2**
> >
> > 4. Selection of the window size is an important point and not only for TNC, but also for similar baselines such as CPC and triplet loss. Overall, the window size should be selected such that it is long enough to contain information about the underlying state, and not too long to span over multiple underlying states. In our settings, we have selected the window sizes based on our prior knowledge of the signals. For instance, in the case of an ECG signal, the selected window size is equivalent to 7 seconds of recording, which is small enough such that the ECG remains in a stable state, and yet has enough information to determine that state. As suggested, we have performed experiments to show the impact of window size on the downstream classification performance (accuracy) of different baselines in the simulated setting. We can see how performance drops for all the methods when the window size is too small or too large. We added this discussion with a full set of results to the appendix of our paper.
> >
> > $\delta$     $\hspace{16pt}$ | $\hspace{24pt}$ 10 $\hspace{24pt}$ |  $\hspace{24pt}$  50 $\hspace{24pt}$   |  $\hspace{24pt}$ 100
> >
> >
> >
> > ----------------------------------------------------------
> >
> > TNC   $\hspace{6pt}$   | $\hspace{6pt}$ 71.60 $\pm$ 0.59 $\hspace{6pt}$ |$\hspace{6pt}$ 97.52 $\pm$ 0.13 $\hspace{6pt}$ | $\hspace{6pt}$ 84.25 $\pm$ 9.08
> >
> >
> >
> > CPC   $\hspace{6pt}$   | $\hspace{6pt}$  51.85 $\pm$ 1.81 $\hspace{6pt}$  | $\hspace{6pt}$ 70.26 $\pm$ 6.48 $\hspace{6pt}$ | $\hspace{6pt}$ 56.65 $\pm$ 0.81
> >
> >
> >
> > T-Loss |$\hspace{6pt}$ 56.70 $\pm$ 1.07 $\hspace{6pt}$ | $\hspace{6pt}$ 76.66 $\pm$ 1.14 $\hspace{6pt}$ | $\hspace{6pt}$ 73.29 $\pm$ 1.58
> >
> > 5. Thank you for pointing out these points for clarification. Here we present some of the answers. We have updated the paper accordingly.
> > Time series are complex in the sense that the data contains temporal dependencies and the underlying generative process of the signals changes over time. Such properties distinguish time series from other data types like images.
> > By robust representations here we mean representations that do not change with small shifts in time, implying time-invariance properties. Instead of just saying robustness, we have now included this full explanation in the paper.
> > The reviewer is correct to point out that “applicable across multiple tasks” is a more accurate term than transferability. We used suggested phrasing in the updated draft to improve clarity and precision.
> >
> > 6. Thank you for noticing the typos. We have updated all of them in the draft.

---

### Official Review · AnonReviewer3 · 2020-10-29
**An advanced HMM**

**Rating:** 6
**Confidence:** 4

**Review:**

The authors defines the notion of temporal neighborhood to segment time series.
It is a location in real vector space where neighboring segments in the original time serie are projected into as well as similar but distant (in time) segments.
The model is trained using a discriminative loss that pushes away (in the feature space) windows distant in time while caring for the possible periodicity [Elkan & Noto 2018]

Overall the paper is well written and easy to follow.
The idea is very interesting and I see several potential consequences and use-cases.
Nevertheless, I have several concerns.

1/ I think the paper lacks classic baselines.
I consider that HMM is part of unsupervised representation learning.
In its original form it does not include NN but, if needed several works have studied variants, eg.:
Firoiu, L., & Cohen, P. R. (2002, July). Segmenting time series with a hybrid neural networks-hidden Markov model.
The authors rightfully raised the issue of how large (in time) a neighborhood is (\eta). For that, Hidden semi-Markov models constitute a nice "dynamic" approach.
As well, HMMs are used to create the synthetic data-set.
* Ok I see, interesting results. Thanks.

2/ Regarding the PU methods, why choosing solution 2 over 1?
* Thanks for the answer. My question was suggesting that it should better justified in the text.

3/ "For an ideal encoder that preserves the neighborhood properties in the encoding space, p(Z_l ∈ N_t ) should
be close to p(W_l ∈ N_t ), where Z_l is the representation of W_l"
I understood that N_t is in the encoding space. So the same letter can not be used for W_l.
I don't see how the training steers the encoder to satisfy this property.
On the other hand, this constrain contradicts the PU discussion.
* Ok I understand better. thanks for the answer.

4/ Figure 1 is not very clear. The encoder outputs Z_t and P(Z_t)

5/ The same network is used for all the baseline. However, Franceschi et al.'s work clearly relies on convolutional layers. Hence not using such is a bit unfair toward their method (synthetic data and HAR).
* I understand your point. Nevertheless some models are very depend on the architecture, since the depend on some feature that not all architecture can provide. And I think your results show that.

6/ Evaluation: Trajectory. Figure 3. Very interesting plot that clearly shows the benefit of this work in terms of interpretability. What happens for a 3 dimensional encoding?
* Here it is my fault, I meant 4 dimensional encoding: as much as the number of hidden states. The point was to see if the model is would learn clearly different representation for each state.

7/ The notion of  temporal neighborhood depends on \eta and is based on a Gaussian distribution around the current timestamp. How does it behave for small and large eta?
A model with a small eta might detect too many segments.
If large, I guess, the models smooths too much the variations and fails to cluster different behavior. Especially it will fail to detect abrupt changes.
In any case, such an analysis is missing.
* Thank you for the explanations.

I have spotted two typo in the last paragraph of page 3:
Note that the Discrimin[a]tor...
...the latent space, since [it] allows...


* I update my score.

---

> ### Author Response · Authors · 2020-11-21
> **Thank you for your constructive feedback**
>
> We would like to thank the reviewer for the thorough review and helpful feedback. We have tried to incorporate all suggestions to improve the clarity of the paper. Our responses to the questions raised are presented below.
>
> 1. In this paper, we have focused our experiments on comparing model-agnostic frameworks for unsupervised representation learning. As pointed out by the reviewer, in the original form, HMMs do not support learning representation for windows of time series, therefore have a somewhat different setting than our baselines. The reviewer suggested a hybrid HMM/neural network to adjust for this, but since our datasets are small and have continuous observations that can be very high dimensional (561 in the HAR dataset), we were not able to get good results from this solution. However, as another experiment, we trained a Gaussian HMM on the HAR dataset and measured the downstream classification performance, presented in the table below. The performance is lower than most baselines mainly because 1) we only have 30 individuals in our dataset; 2) the observation distribution is modeled as a Gaussian with diagonal covariance since the number of training samples is not enough to learn the full covariance; 3) the representations correspond to a single observation as opposed to a window.
>
> $\hspace{48pt}$ | $\hspace{3pt}$ Accuracy $\hspace{3pt}$ |  $\hspace{5pt}$ AUPRC  $\hspace{5pt}$ | $\hspace{5pt}$AUROC
>
> HAR dataset | $\hspace{10pt}$  48.74 $\hspace{10pt}$ |$\hspace{10pt}$ 0.683 $\hspace{10pt}$| $\hspace{10pt}$ 0.356
>
> 2. We have used the second approach because the first method requires finding possible negative data from the unlabeled set, therefore, it relies heavily on heuristics for identifying those samples [Kiryo et al., 2017]. In our setting, this can be very challenging because of our lack of knowledge about the distribution of the underlying states. In contrast, the second approach relies on estimating the probability of having positive samples in the unlabeled set, which can be set and tuned in the learning process.
>
> 3. We realize the notation used can cause confusion, and we apologize. We updated the draft and the notations for more clarity. To summarize, the discriminator predicts the probability of 2 encodings belonging to the same neighborhood in time. Using the loss function defined in equation 1, we directly maximize the predicted probability for pairs of encodings from the same neighborhood while minimizing it for non-neighboring samples.
> The point raised about the PU assumption is very important. It is
>  exactly why we reweight our cross-entropy to adjust for windows from outside the neighborhood that is expected to have a low probability but are in fact similar to W_t.
>
> 4. We realize the visualization of this concept was confusing and as suggested by the reviewer, we have updated the figure for a more simple and accurate representation of the concept.
>
> 5. The goal of our evaluations was to obtain relative performance of the representational frameworks regardless of the architecture (hence we had to keep the architectures the same), rather than obtaining the best overall performance for each dataset.
> Different problems and data types will require different model architectures, so we focus on model-agnostic representational frameworks.
>
> 6. This is an interesting question. We have now included the trajectory plot for lower dimensional encoding to the appendix. This figure shows the trajectory of a simulated signal, with 4 underlying hidden states and 3-dimensional representations. As expected, the lower dimensionality reduces the clustering performance of the algorithm. While it still distinguishes some of the states, some others end up being grouped together. This is interesting and in retrospect is not unexpected, as the set of the underlying generative processes for blue and yellow regions are the same and the difference is in which features follow which process. This tells us that lower dimensionality captures a more general trend across signals, not having the capacity to capture feature level distinctions.
>
> 7. The size of the neighborhood and its parameterization are very important. In fact, this is precisely the reason why TNC uses a statistical test like ADF to determine the optimal $\eta$ parameter for the neighborhood. Small neighborhoods result in neighboring samples with high overlap. In such a situation, what’s likely to be encoded in the representation is the overlap between the windows rather than anything informative. On the other hand, if $\eta$ is too large, the neighborhood will span over multiple underlying states. Therefore some of the samples from the same neighborhood will have different underlying states. This demonstrates the importance of an automated approach for the estimation of the neighborhood size. We have added this discussion to the draft.
>
> 8. Thank you for noticing the typos, we have updated the draft to fix them.

---

### Official Review · AnonReviewer1 · 2020-11-04
**Has potential to be a good paper with edits**

**Rating:** 6
**Confidence:** 4

**Review:**

Summary

The paper proposes an unsupervised representation (embedding) learning method for time-series. While unsupervised representation learning has been extensively studied and shown good performance in fields like NLP and vision, it is relatively new to the time-series community. This paper, in contrast to recent work (CPC and Triplet-Loss), has the following differences:
1. It estimates stationary temporal windows using statistical testing for stationarity/non-stationarity.
2. It learns embedding using contrastive learning as in CPC and Triplet-loss, but additionally takes into account that fact that naive negative sampling may include false negatives and hurt embedding learning on time-series that have strong seasonality. Instead, it adopts the framework Positive Unlabeled learning to address this issue.

Quality and clarity

I believe the work itself is of good quality (or has the potential to be), but the presentation of the work is not very clear to me:
1. The neural network architecture adopted is not well explained: if the architecture varies across problems then the authors should clearly state this and provide more explanation on the general framework using Figure 1; currently, the framework in Figure 1 is also not well discussed.
2. The way how the statistical test (Augmented Dickey Fuller) is incorporated for window-detecting is not clear. According to the explanation in Section 2, "For every W_t, we gradually increase the neighborhood size and measure the p-value from the test", it seems the windows will have varying sizes according to test p-values; but in all experiments, the window sizes are fixed across the entire time-series. The authors should provide some explanation to this.


Originality

The paper builds on existing framework: contrastive learning based unsupervised representation learning, but proposes a new method to cater to the peculiarity of long-term time-series (non-stationarity and seasonality).

Significance

Given that exploration of unsupervised representation learning in time-series is at early stage, I believe this work will inspire future works along this direction.

---

> ### Author Response · Authors · 2020-11-21
> **Thank you for your constructive feedback**
>
> We would like to thank the reviewer for the positive feedback. We are happy to clarify our manuscript in response to the reviewer's questions.
>
> 1. The goal in our experiments was to compare the performance of different learning frameworks, independent of the encoder architecture, and for that purpose, we used the same encoder model across different baselines for each experiment setup. We wanted to see how each framework can use the capacity of a simple encoder model to learn meaningful representations, which is the reason behind choosing a simple RNN encoder for the first 2 experiments. For the ECG data, however, a single layer RNN was too simple to model the high-frequency ECG signals, so we had to use a more complex architecture that fits the data better. The architecture that is used is inspired by state-of-the-art models for ECG classification problems. A summary of these architectures can be found in (https://github.com/hsd1503/DL-ECG-Review). We have added this discussion to the paper for clarification. In addition, as also suggested by Reviewer 2, we have simplified Figure 1 and explained the framework more thoroughly using this visualization.
>
> 2. We would like to clarify that the ADF test intends to find the neighborhood range (governed by $\eta$) and is separate from the window size $\delta$. For every $W_t$, we want to find the neighborhood range around that window that indicates a stationary region. For that purpose, we gradually increase the “neighborhood size $\eta$” and measure the $p$-value using the test. This way we find the widest neighborhood within which the signal remains relatively stationary. We have added more information to the draft to clarify this point.

---

### Decision · Program_Chairs · 2021-01-07
**Final Decision**

**Decision:**

Accept (Poster)

**Comment:**

The paper proposes an unsupervised representation (embedding) learning method for time-series. Overall, the paper is well-motivated, well-written and easy to follow. As agreed by all reviewers, the idea is interesting. To further improve the paper, the authors are encouraged to justify the choice of encoder architectures and window size, and describe more clearly how the statistical test is incorporated.